# A high-throughput, 28-day, microfluidic model of gingival tissue inflammation and recovery

Ashley L. Gard[1,3], Rebeccah J. Luu[1,3], Ryan Maloney[1], Madeline H. Cooper[1], Brian P. Cain [1],
Hesham Azizgolshani[1], Brett C. Isenberg[1], Jeffrey T. Borenstein [1], Jane Ong[2], Joseph L. Charest[1] &
Else M. Vedula [1]✉

Nearly half of American adults suffer from gum disease, including mild inflammation of gingival tissue, known as gingivitis. Currently, advances in therapeutic treatments are hampered by a lack of mechanistic understanding of disease progression in physiologically relevant vascularized tissues. To address this, we present a high-throughput microfluidic organ-on-chip model of human gingival tissue containing keratinocytes, fibroblast and endothelial cells. We show the triculture model exhibits physiological tissue structure, mucosal barrier formation, and protein biomarker expression and secretion over several weeks. Through inflammatory cytokine administration, we demonstrate the induction of inflammation measured by changes in barrier function and cytokine secretion. These states of inflammation are induced at various time points within a stable culture window, providing a robust platform for evaluation of therapeutic agents. These data reveal that the administration of specific small molecule inhibitors mitigates the inflammatory response and enables tissue recovery, providing an opportunity for identification of new therapeutic targets for gum disease with the potential to facilitate relevant preclinical drug efficacy and toxicity testing.

[1] Bioengineering Division, Draper, Cambridge, MA, USA. [2] Colgate-Palmolive Company, Piscataway, NJ, USA. [3] These authors contributed equally: Ashley L. Gard, Rebeccah J. Luu. ✉email: evedula@draper.com

Gum tissue serves as the primary protective barrier and physical defense in the mouth against bacterial invasion into gingival tissue and the bloodstream. Half of the American adults suffer from periodontal disease[1], and the World Health Organization reports that gum disease remains a very important global health burden[2]. In the early stages of gum disease, microbial plaques accumulate and form biofilms, eliciting inflammatory-mediated gingivitis, a preventable and reversible infection of the gingival tissue, and alteration of the protective barrier[3]. The oral disease research community remains limited by the tools and technology available to probe gum disease pathophysiology, and its broader consequences to systemic health and disease. This gap in laboratory research tools has hampered advances in understanding oral inflammation at a cellular level, as well as the ability to evaluate oral therapeutic and prophylactic agents in vitro.

Microphysiological systems (MPS) and organs-on-chip (OOC)[4–8] are engineered, realistic, complex, and human-predictive in vitro models of tissue to aid preclinical development by allowing the study of mechanisms of health and disease, and screening therapeutic candidates for the treatment of various pathophysiological states. MPS have advanced rapidly toward applications in disease and toxicity research for the liver, kidney, lung, gut, vasculature, brain, and various interconnected organ models[9–13], while other tissues, such as oral tissues, remain less developed. Early examples of in vitro modeling of the oral mucosa are often monoculture models (e.g., gingival epithelia), and may lack one or more relevant biochemical or biophysical cues, such as extracellular matrix (ECM), fluid flow, or an air-liquid interface (ALI)[14–18]. Microfluidic devices and perfusion systems have existed within the in vitro tissue model and MPS field over the last decade[6,19–21], though these technologies and capabilities are only beginning to find applications in the dental, oral and craniofacial research communities. Further, many MPS are low-throughput and do not provide extended culture times and many of the biophysical cues relevant to mimicking oral gingival physiology have not yet been integrated into these systems[22]. Recent reports of dynamic in vitro models of the human gingiva are beginning to incorporate biophysical cues, including bioreactors that support perfusion through a collagen sponge to model the periodontal pocket[23], PDMS scaffold platforms supporting an indirect gingival co-culture[24], and an oral mucosa-on-chip with the surface tension-driven flow for up to 7 days[25]. An in vitro model incorporating a multi-cell type, multi-layered tissue in a high-throughput model with programmable flow control and integrated functional metrics would further enable the study of oral gingiva in both academic and commercial settings.

Here, we present a high-throughput Microfluidic model of Oral physiology for Understanding Tissue Health (MOUTH) to model healthy and disease states of the gum tissue in vitro. The MOUTH model is a multi-layered, multi-phenotypic gingival tissue in a microfluidic platform. Human primary gingival cells grown in co-culture with human microvascular endothelial cells form an oral-to-systemic barrier for up to 4 weeks and allow characterization of barrier function, protein expression, and cytokine secretion. The MOUTH model has been established in a 96-device microfluidic membrane bilayer platform, known as PREDICT96[26], that has been employed to support other in vitro mucosal and vascular tissue models and builds off of previous platform technologies[7,8,13,27,28]. We established a multi-week experimental window in which transepithelial electrical resistance (TEER) was stable, indicating mature, homeostatic barrier function over an extended length of time in vitro. Furthermore, we demonstrated an inflammatory disease state of MOUTH, in which cytokine secretion increased and barrier function decreased in response to an initial inflammatory stimulus. Tissue recovered upon removal of the stimulus and protection with a small-molecule inhibitor was demonstrated. The multi-week culture window and high-throughput nature of the model enabled statistically significant evaluation of multiple dosing schemes of stimuli in devices. These results demonstrate that MOUTH provides a robust model for assaying the gingival tissue's physiological state and response to an inflammatory stimulus over long culture times for applications in oral product and therapeutic development and disease modeling.

## Results

**Establishing a microfluidic model of gingival barrier tissue.** The MOUTH triculture model consisted of three cell phenotypes configured within a high-throughput OOC platform technology. The platform technology supports heterotypic cell type complexity in high-throughput (plate: Fig. 1a) with programmable flow control (pump lid: Fig. 1b) and integrated TEER sensing (plot: Fig. 1d)[29]. Human oral keratinocytes (hOKs) differentiated and formed a multi-layer tissue barrier along with donor-matched human gingival fibroblasts (hGFs) and human microvascular endothelial cells (hMVECs) for up to 32 days under unidirectional, recirculating fluid flow. A healthy MOUTH model was characterized by multi-cellular and multi-layered tissue constructs as seen by immunofluorescent (IF) imaging, barrier function through 30 days of culture, and cytokine secretion. Endpoint immunofluorescence microscopy confirmed a confluent layer of hOKs (Fig. 1c, top channel) and hMVECs (Fig. 1c, bottom channel), and sporadic presence of hGF in the basal tissue layers (Supplementary Fig. 5). A multi-layered structure of hOKs, seen in confocal cross sections (Fig. 1e), was measured up to 200 μm thick and consisted of five to eight layers of cells. Evaluation of vascular endothelial cell (von Willebrand Factor, Vimentin) markers and viability (Calcein) using fluorescent imaging showed a viable and confluent hMVEC monolayer at the end of the 4-week culture (Fig. 1f, g). The differentiated epithelium comprised a thin layer of large hOKs towards the apical side of the tissue with granules visible throughout the cell bodies (Fig. 1h), an intermediate spinous layer of polyhedral hOKs with larger nuclei and numerous intercellular connections (Fig. 1i), and one to two basal layers of compact hOKs with positive expression of cytokeratin 14 (Ck14) (Fig. 1j),.

Across multiple experiments, TEER, which served as a main metric of tissue health, increased over the first 10 days of culture, at which point the barrier function typically reached a plateau for the final 20 days of the experiment (Fig. 1d). The average barrier function of the MOUTH model over a total of 288 replicates across three experiments between days 10–30 was $317\,\Omega\,cm^2$, with a standard deviation between all replicates of $49\,\Omega\,cm^2$. Approximately 1 week into the plateau period, most experiments exhibited a 25% dip in TEER for 3–4 days before returning to or exceeding the baseline plateau value of TEER. A second main metric of tissue health was the secretion of PGE2, a lipid arachidonic acid-derived prostaglandin eicosanoid, which is known to help mediate the inflammatory response and has been shown to be positively correlated with the severity of periodontal disease[30]. In one experiment, PGE2, monitored on the first day of a plateau (Day 11) and the last day of culture (Day 32) was undetectable at both time points. In the same experiment, to assess the health of the constructed tissue, we examined a profile of 18 inflammatory biomarkers (Supplementary Fig. 1) from the basal media on day 11 and day 32. We found that none of the inflammatory biomarkers increased significantly over the course of the experiment, and that only one, MIP-3a, increased even slightly. All other biomarkers either maintained their concentration or decreased over time, while TEER measurements indicated

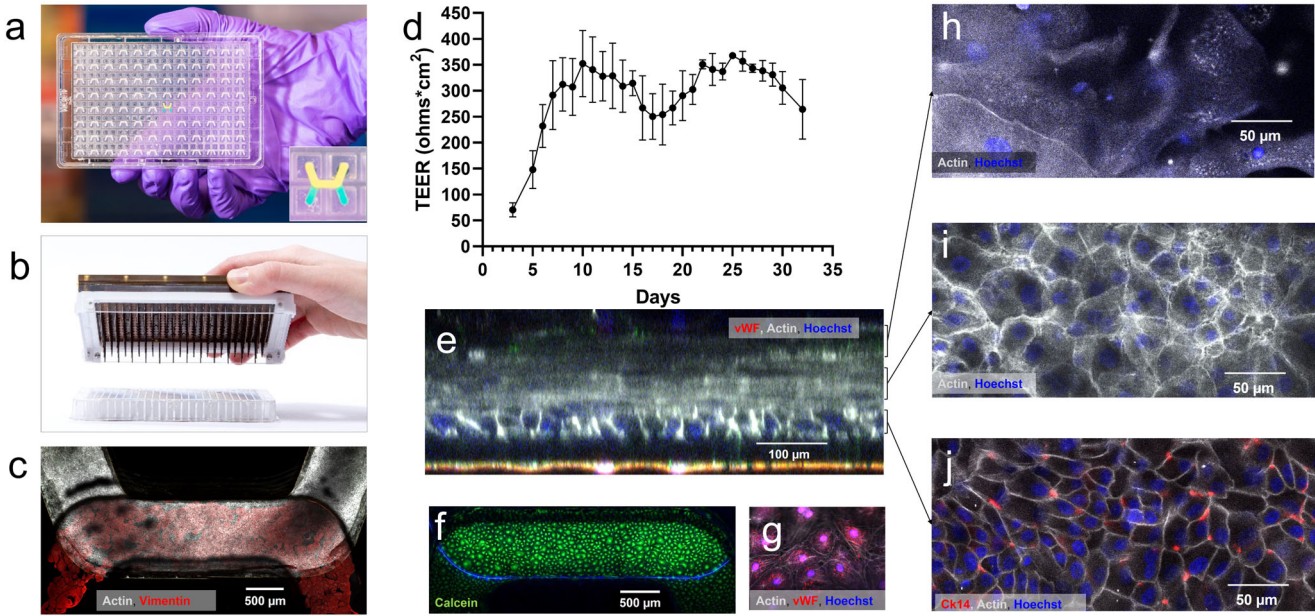

**Fig. 1 A triculture model of human gingival tissue, comprised of human gingival fibroblasts (hGFs), human oral keratinocytes (hOKs), and human microvascular cells (hMVECs) have been cultured in PREDICT96 for 1 month under recirculating fluid flow. a** Draper's high-throughput microfluidic culture platform contained 96 microfluidic devices with a top and bottom channel (inset) separated by a porous membrane (1 μm pore size, 1e6 porosity). **b** The microfluidic-based pump exists in the lid of the plate and delivers independent fluid flow to the top and bottom channels of each microfluidic device. **c** A maximum intensity projection of the top channel of a single PREDICT96 device showed a confluent layer of hOKs after 32 days of culture. **d** Mean TEER measurements from three separate experiments and up to 288 replicates per day indicated a general trend of barrier formation over the experimental duration. The tissue barrier increased over the first 10 days of culture as layers proliferated and junctions formed, reaching a plateau sustained between 250–360 Ohms cm$^2$ for the following ~21 days. Error bars represent standard deviation. **e** A confocal orthogonal view of the MOUTH tissue showed >100 μm thick hOK tissue on the top side of the membrane. **f** Calcein live stain of hMVEC layer at day 32 demonstrates a robust endothelial layer can be maintained through extensive culture with **g** von Willebrand factor (red), indicating vascular-specific staining on the bottom side of the membrane. **h–j** Changes in tissue morphology and structure could be seen at different z-planes of the tissue, including **h** a layer with visible granules in the upper z-plane, **i** a distinct actin structure in the middle z-plane, and **j** a basal proliferative layer of hOKs.

the total barrier function remained steady. This is especially true for MCP-1, GROα, IL-8, and VEGF, which more than halved from day 11 to day 32, indicating an improving baseline of inflammatory markers.

*Inflammation states in MOUTH.* Stimulation by an inflammatory trigger resulted in altered TEER and increased levels of secreted inflammatory markers in the MOUTH model. A clinically relevant, pathophysiological cocktail of 300 ng/ml each of TNF-α and IL-1β (Stim) was dosed for 24 h to serve as the inflammatory trigger, after which TEER was measured daily for the duration of the experiment and inflammatory biomarkers were analyzed over a 48 h window (Fig. 2a). Secreted levels of PGE2 measured in the bottom channel of stimulated devices increased more than sixfold compared to vehicle control devices within 24 h, and remained more than fivefold higher than vehicle controls for at least 48 h (Fig. 2b). Inflammatory cytokines, MIP-3α, IL-10, and IFN-γ were all significantly increased at 24 and 48 h post-stimulation compared to untreated and vehicle controls (Fig. 2c–e), indicating a multi-factor response to the inflammatory trigger. The overlapping data points in Fig. 2c–e represent a coefficient of variation of less than 10% for almost all datasets. A Luminex panel of an additional 15 cytokines and chemokines demonstrates an additional impact of inflammation (Supplementary Figs. 7, 8), in which many were upregulated following inflammatory stimulation, including CSF, G-CSF, RANTES, VEGF, IL-33, IP-10, IL-6, GROα, IL-8, and IL-4. MCP-1, G-CSF, PDGF-aa, and PDGF-ab/bb expression levels, on the other hand, were not significantly affected by IL-1β and TNF-α stimulation.

Over the course of the 6 days following stimulation, vehicle control TEER generally stayed consistent, while the TEER of stimulated devices decreased for 4 days before reaching a plateau (Supplementary Fig. 2a). The same data shown over 48 h in a bar plot highlights the TEER values of stimulated devices had decreased 14% by 48 h post-stimulation, while the vehicle-treated (0.1% BSA) tissues remained relatively unchanged (Supplementary Fig. 2b).

The time at which inflammatory triggers were introduced to the tissue had an effect on inflammation response. We stimulated separate devices at early and late time points to investigate differences in their tissue response. Summarized in Fig. 3a, "Early Stim" denotes devices stimulated on Day 10 when the tissues first reached a TEER plateau and "Late Stim" denotes devices stimulated 3 weeks into the culture at Day 21. Early Stim and Late Stim devices were separate replicates, such that the Late Stim devices were not previously stimulated. Average secretion levels of PGE2 trended similarly in both Early and Late Stim devices (Fig. 3b), but was only statistically significant (more than twofold) at the Early Stim compared to vehicle control after 48 h. The initial pre-inflammatory stimulation baseline levels of PGE2 showed no statistical difference when compared to one another at time point 0. MIP-3α and IFN-γ secretion trends were similar between Early and Late Stim, as both were higher compared to controls, though Late Vehicle devices exhibited higher variability, rendering only the Early Stim statistically significant at both 24- and 48 h post-stimulation. IL-10 secretion significantly increased in response to an earlier dose of a stimulant but exhibited a more variable response to the later

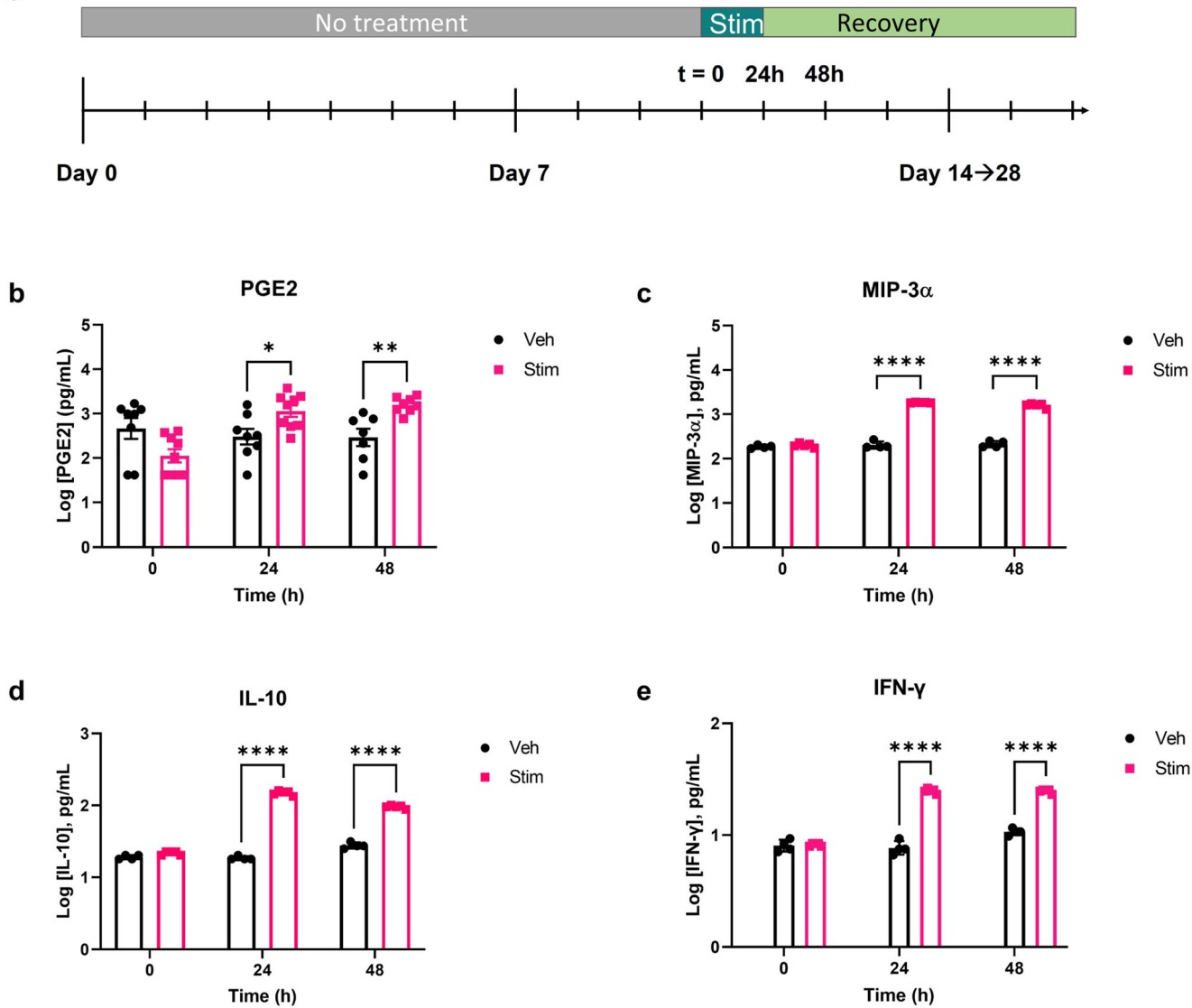

**Fig. 2 MOUTH tissue responded to a single dose of inflammatory stimuli. a** MOUTH samples were stimulated with an inflammatory trigger of 300 ng/mL each of TNF-α and IL-1β (Stim) around day 10 of culture and evaluated compared to vehicle controls. **b** Secreted levels of PGE2 increased in bottom channels of MOUTH Stim devices compared to vehicle control devices within 24 h ($p < 0.0332$) and remained higher than vehicle controls for at least 48 h ($p < 0.0021$). Data from **b** are averaged across three experiments ($N = 3$–12 per experiment). **c**–**e** Devices stimulated with inflammatory cytokines IL-1β + TNF-α produced an increase in MIP-3α, IL-10, and IFN-γ production at 24 and 48 h following stimulation. Data from **c**–**e** are from one experiment with $N = 3$. In all cases, error bars represent standard deviation. Significance was determined by Tukey's test. Significance: $^*p < 0.0332$, $^{**}p < 0.0021$, $^{***}p < 0.0002$, $^{****}p < 0.0001$.

dose (Fig. 3c–e). A more extensive panel of inflammatory cytokines reveals a deeper look into tissue response at both early and late stimulation time points (Supplementary Fig. 8).

By 6 days post-stimulation, the TEER of Late Stim devices dropped by twofold, similar to the 0.41-fold drop of Early Stim devices over the same time frame (Supplementary Fig. 3a). The barrier function of Early Stim devices dropped 0.15-fold within 48 h, whereas the Late Stim devices did not lose a significant amount of barrier function in the same time frame. In fact, late Stim TEER increased temporarily (1.23- and 1.27-fold) before decreasing and matching the Early Stim TEER response trend (Supplementary Fig. 3b).

**Treating inflammation in MOUTH.** The MOUTH model inflammatory response can be modulated by small-molecule inhibitors, a representative timeline of which can be seen in Fig. 4a. Both barrier function and cytokine secretion, our main metrics of inflammation, were reduced when dosing MOUTH with a p38 mitogen-activated protein kinase (MAPK) inhibitor SB203580 (SB). After 24 h, PGE2 release from SB + Stim devices was 0.84-fold lower than Stim devices. These same trends were seen at 48 h post-stimulation as well as in devices that only received SB as a control (Fig. 4b). Only IL-10 secretion was significantly reduced in SB + Stim devices when compared to Stim devices, indicating that SB had the strongest effect on reducing PGE2 and IL-10 secretion compared to MIP-3a and IFN-γ secretion (Fig. 4c–e). Secretory levels of Fractalkine, GM-CSF, G-CSF, RANTES, VEGF, IL-33, IP-10, and IL-6 increased following inflammatory stimulation, a result that was inhibited when pre-treated with small-molecule SB within at least 48 h. GROα, IL-8, and IL-4 were also upregulated following inflammation stimulation, but not protected when pre-treated with SB.

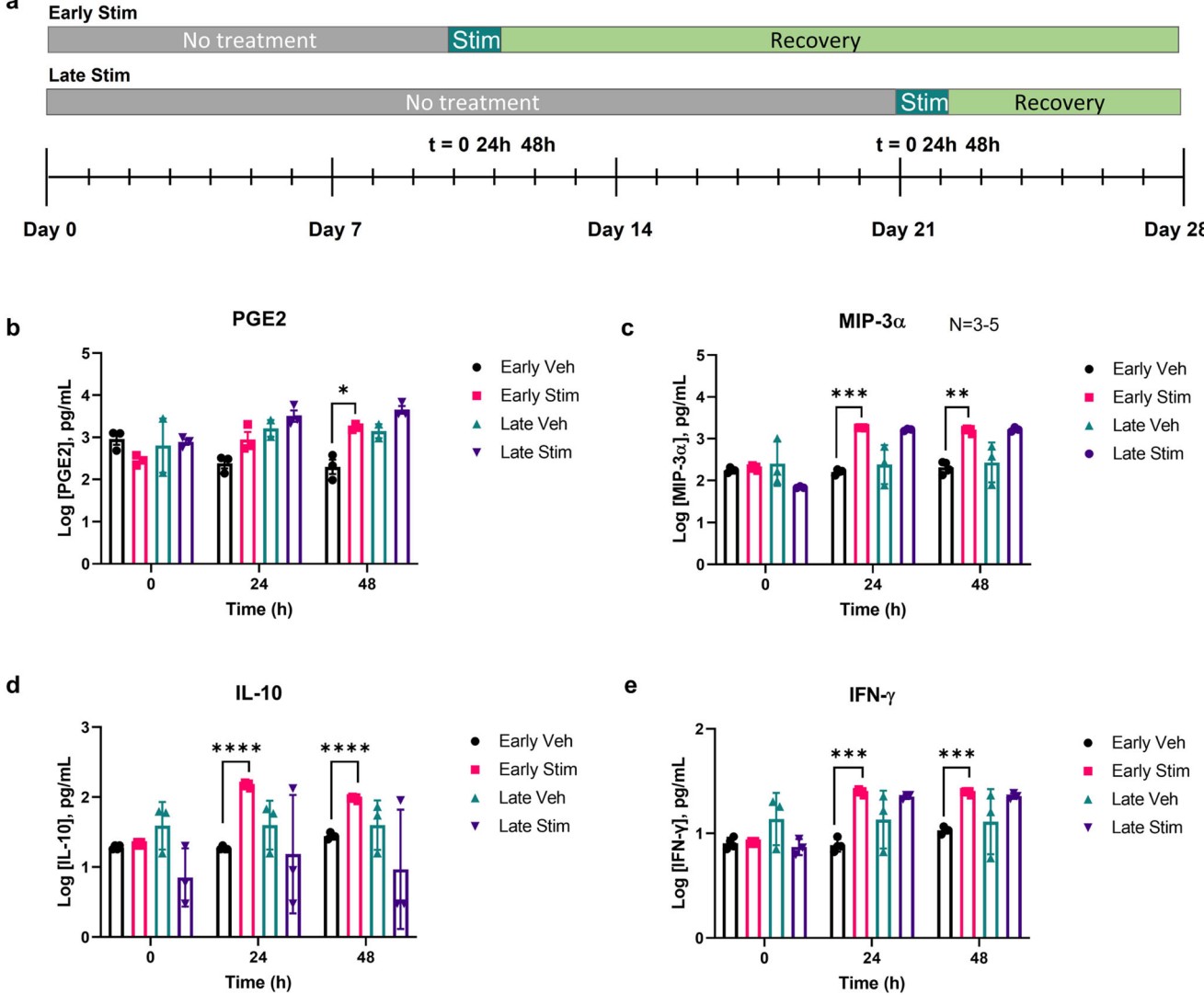

**Fig. 3 MOUTH tissue responded to inflammatory stimuli at different time points over the 20-day testing window. a** An early and late stimulation were applied to MOUTH devices at 10 and 21 days, respectively, and evaluated compared to vehicle controls. **b** Similar trends of secreted levels of PGE2 were observed between Early and Late Stim conditions compared to corresponding vehicles, but only Early Stim was significant by 48 h compared to controls. Data represent at least three replicates from one experiment. **c–e** MOUTH tissue has a similar inflammatory response to stimulation during the testing window. Devices stimulated with inflammatory cytokines TNF-α + IL-1β presented significant increases in MIP-3α, IL-10, and IFN-γ at 24 h under Early Stim conditions, and trended similarly in MIP-3α and IFN-γ secretion under Late Stim conditions. In all cases, error bars represent standard deviation. Significance was determined by Tukey's test for **b**, and Dunnett's T3 multiple comparisons test for **c–e**. Significance: $*p < 0.0332$, $**p < 0.0021$, $***p < 0.0002$, $****p < 0.0001$.

Devices treated only with SB for 1–2 h, maintained barrier function for 48 h compared to devices that were stimulated with IL-1β and TNF-α without inhibitor (Stim), which experienced a significant decrease in TEER compared to untreated and vehicle controls by 48 h post-stimulation (Supplementary Fig. 4a). However, SB had only a temporary protective effect on TEER, as devices pre-treated with SB followed by stimulation with inflammatory triggers (SB + Stim) lost barrier function by 4 days post-stimulation to match TEER values of Stim devices (Supplementary Fig. 4b). In the same experimental window, devices treated with SB-only exhibited TEER trends similar to control and vehicle devices, indicating that SB alone did not disrupt barrier function.

Sequential dosing of the same device with inflammatory stimuli and/or inhibitors demonstrated the effects of chronic dosing, as illustrated in a testing timeline (Fig. 5a). SB protected the barrier integrity of MOUTH tissue for 48 h after the first dose, and at

least 6 days after the second dose. Devices received the first dose of TNF-α and IL-1β with or without SB on Day 11 ("Stim 1x") and the second dose on Day 21 ("Stim 2x"). An extended TEER timeline (Fig. 5b) illustrates that by 3 days post-Stim 1x, TEER decreased by 31% until it stabilized on day 14 for several days. On day 17, TEER began to recover until hitting another plateau on Day 21, recovering to 82% of the pre-stimulation TEER value. Devices were then subjected to a second dose of TNF-α and IL-1β (Stim 2x). Barrier function, in this case, did not recover after Stim 2x and TEER continued to decrease from 3 days post-stimulation and for the following 5 days until the end of the experiment. The SB inhibitor again had a brief protective effect on TEER in SB + Stim devices in the first 48 h post-Stim 1x, but TEER eventually matched and followed the trend of Stim conditions. SB treatment on the second dose of stimulation (Pre-SB + Stim 2x), however, appeared to maintain TEER values for the remaining 6 days of the experiment. For both Stim 1x and Stim 2x, vehicle

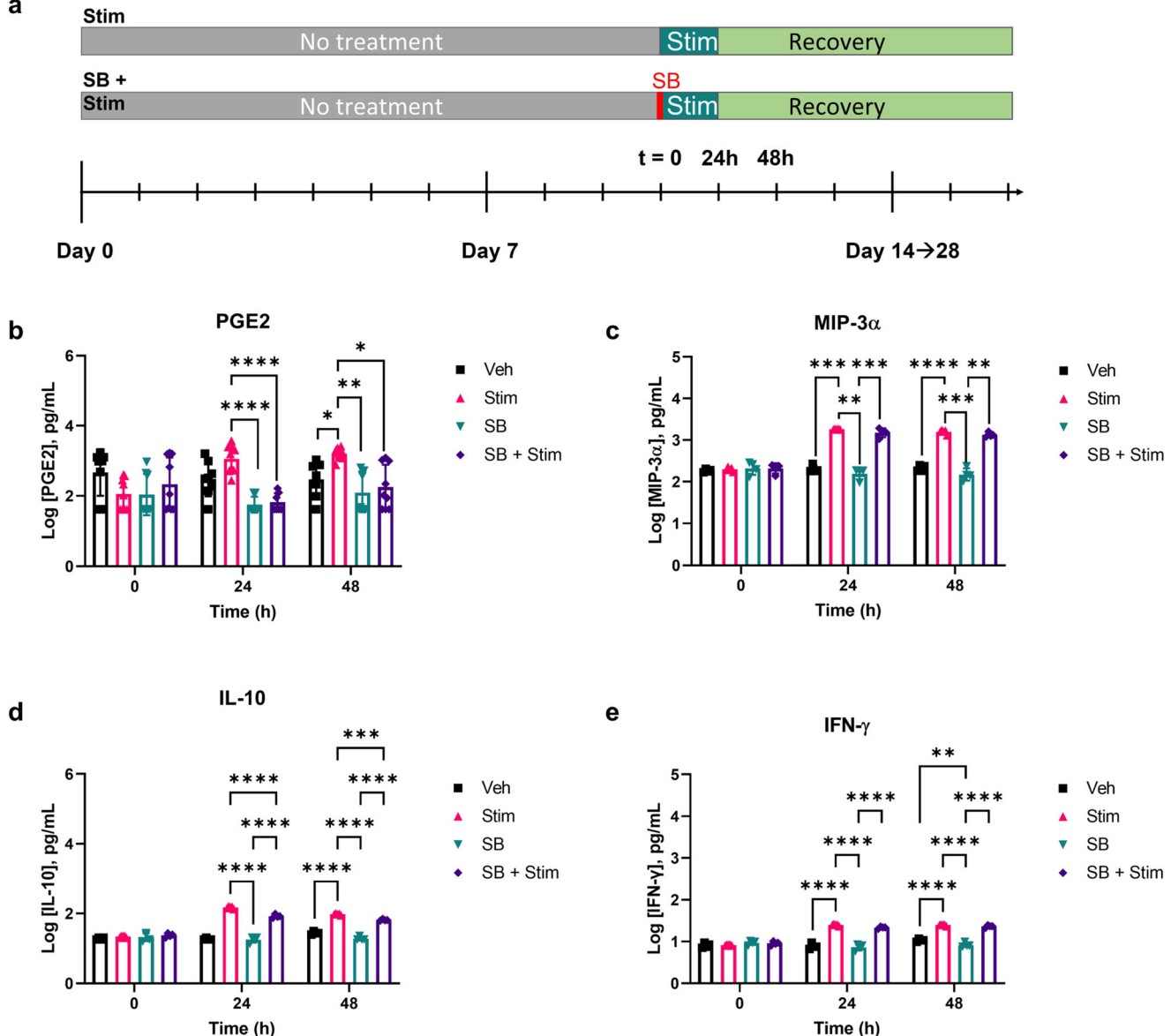

**Fig. 4 MOUTH inflammatory response was modified with a small-molecule inhibitor. a** Devices were treated with a MAPK inhibitor (SB) 2 h prior to stimulating with inflammatory triggers to evaluate the modulation of tissue response compared to controls. **b** Twenty-four and 48 h post-stimulation, secreted PGE2 levels in both SB and SB + stim devices were significantly lower than Stim devices. The data presented is averaged across three experiments ($N = 3$–6 per experiment). **c–e** Preventative treatment of devices with SB prior to stimulation significantly prevented IL-10 secretion but did not significantly prevent the secretion of MIP-3α or IFN-γ. Devices treated with just SB only decreased IFN-γ secretion relative to vehicle controls. In all cases, error bars represent standard deviation. Significance was determined by Dunnett T3 multiple comparisons test for **b**, **c** and Tukey's test for **d**, **e**. Significance: *$p < 0.0332$, **$p < 0.0021$, ***$p < 0.0002$, ****$p < 0.0001$.

control TEER did not change significantly in the 48 h post-stimulation, indicating a specific response to TNF-α and IL-1β. Taken together, these results indicate that tissue may respond differently to both inflammatory conditions and preventative measures depending on its history of inflammation or additional maturity of the tissue. All devices receiving SB maintained TEER values for the entirety of the experiment compared to controls. For PGE2 secretion (Fig. 5c), significance between Vehicle and Stim devices cannot be established due to the variability in the vehicle control device data. However, the mean of Stim 1x and Stim 2x was about 1 log higher than the Vehicle control devices at 24 h post-stimulation. PGE2 levels on day 21 of Stim 1x devices, which returned to pre-stimulation levels, indicated that tissue had

recovered to baseline (Supplementary Fig. 1). Prophylactic dosing with SB prior to stimulation prevented an increase of PGE2 release in the 24 h after dosing in both Stim 1x and Stim 2x conditions.

## Discussion
Validation using known inflammatory stimuli and clinical metrics of the inflammatory response are crucial to establishing an in vitro gingival tissue model that possesses clinical significance. Preclinical pipelines, in particular, rely on traditional in vitro cell culture or animal models, which each suffer from low predictive accuracy and low throughput, respectively. An

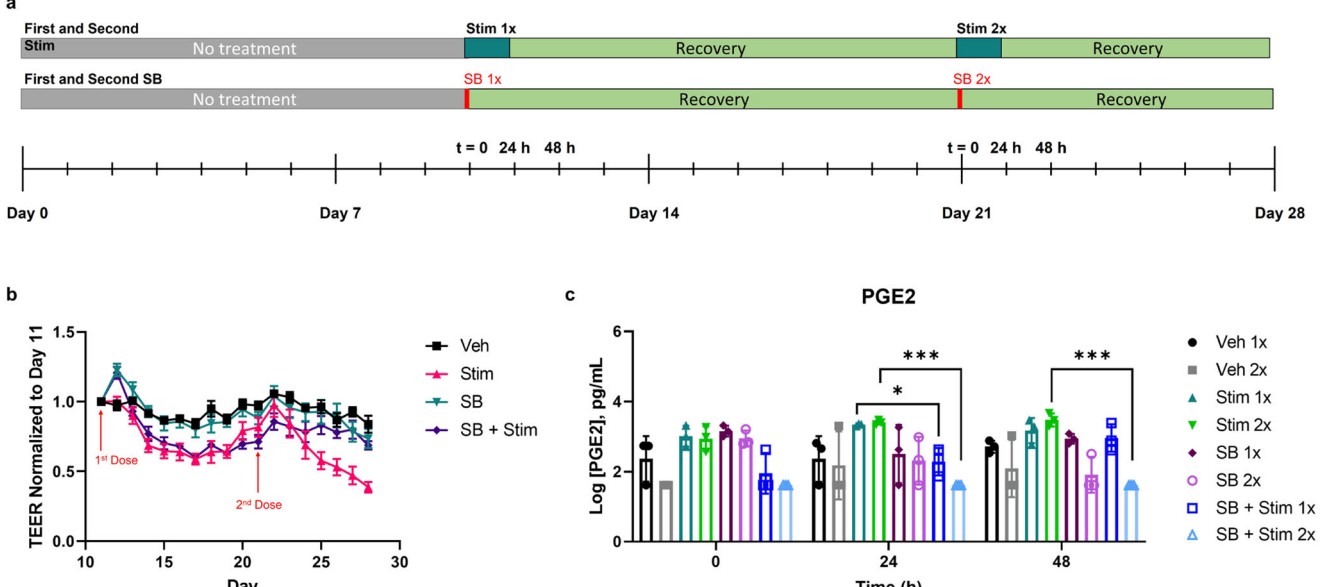

**Fig. 5 MOUTH tissue responded to multiple doses of inflammatory stimuli and/or inhibitors. a** A dosing scheme illustrates consecutive doses of inflammatory stimulation and inhibitory treatment with a MAPK inhibitor (SB) versus controls. **b** Barrier function of Stim devices and Pre-SB + Stim devices trended similarly through the first dose of inflammatory stimulants, in which TEER decreased for several days post-stimulation before increasing and plateauing at sub-baseline levels by Day 21. The first dose of inflammation stimulant administered on Day 11 of culture (Stim 1x) decreased TEER values by 31% within 3 days and recovered to 82% of baseline TEER within 10 days. TEER values of devices pre-treated with SB prior to the first dose of stimulant (Pre-SB + Stim), decreased by 23% within 3 days and recovered to 71% of baseline TEER within 10 days. The trend of the two conditions diverged after the second dose, which was administered 10 days after the first dose. The second dose (Stim 2x), did not significantly reduce TEER in the first 48 h, but ultimately TEER was reduced by 60% by 7 days post-second dose relative to Day 11 values. By the end of culture on Day 28, compared to the average TEER measured on Day 11 before Stim 1x, TEER of Stim devices had decreased by 61%, and TEER of Pre-SB + Stim devices had decreased by 31%. TEER of vehicle control devices had decreased by 16%. **c** Secreted PGE2 levels increased 24 hours after devices were stimulated the first time (Stim 1x). These levels recovered to pre-stimulation levels by the second round of dosing, on day 21, when the same devices were dosed again with IL-1β and TNF-α (Stim 2x). With the second dose, secreted PGE2 levels increased within 24 h, following the same trend as the first dose. SB 2x devices had a decrease in PGE2 release, also similar to the first dose of SB on day 11. Pre-incubation with SB inhibited inflammatory response for 24 h after the first and second inflammatory doses, as indicated by the preservation of PGE2 levels compared to Stim conditions. Data represent at least three replicates from one experiment. In all cases, error bars represent standard deviation. Significance was determined by unpaired Welch's *t*-test. Significance: *$p < 0.0332$, **$p < 0.0021$, ***$p < 0.0002$, ****$p < 0.0001$.

advanced, high-throughput tissue culture platform like MOUTH offers a balance of complex human tissue and throughput in the preclinical space, especially for oral- and self-care production companies that do not use animal models. Furthermore, the month-long functionality of the model provides a platform from which to better study and understand multi-window scenarios such as inflammation response and tissue recovery in the presence of therapeutics, or acute disease progression. Further validation and development of the model would be needed to replace animal models for more chronic disease and toxicity evaluation.

Defining the tissue baseline/homeostatic profile, incorporating relevant controls, characterizing tissue inflammation over time to known stimuli, and having the throughput necessary to perform statistically significant analyses, are vital aspects of the validation process of the MOUTH model. We characterized our tissue baseline using metrics of TEER, cytokine and PGE2 release, and IF imaging and observed consistent trends in barrier function and tissue morphology over a 30-day culture period. Additionally, during the 30-day culture period, we quantified the baseline inflammatory profile of the MOUTH model by multiplexed fluorescent bead-based immunoassay and observed that the prevalence of the analytes surveyed (Supplementary Fig. 1) was consistent with observed inflammatory cytokine and chemokine profiles of gingival cervicular fluid from periodontally healthy human subjects[31–44]. Similar to other in vitro gingival tissue models using both submerged and ALI culture methods[15],

MOUTH tissue morphology maintained five to eight layers of keratinocytes measuring up to 200 μm thick. Whereas other tissue models maintain this structure for up to 7 days, MOUTH maintains its architecture for up to 28 days. A video of MOUTH tissue structure (Supplementary Video 1) demonstrates differentiated morphology as the optical field of view spans nearly 125 μm from the microvascular endothelial cell layer, through the densely packed basal layer of cells and multiple layers of para-keratinized stratified squamous epithelium with evidence of granular and spinous layers. In addition to the gingival-origin of the keratinocytes and fibroblasts, the tissue morphology indicates gingival-specific histology relevant to the applications of MOUTH. Additional features, including rete ridges involved in supporting mastication forces, would be of interest to model in future MOUTH applications using either previously developed membrane surface topography[45] or other engineered microenvironmental features.

hGFs formed a confluent monolayer prior to introducing the hOK. However, after 30 days of triculture conditions, the hGF were often difficult to distinguish from the other two tissue types but could be seen sporadically throughout the basal layers due to their elongated morphology, TE-7 expression, and actin structure (Supplementary Fig. 5). Two additional primary fibroblast cell types were evaluated while establishing the MOUTH triculture model: human oral fibroblasts (hORF) and human dermal fibroblasts (hDF). In the microfluidic microenvironment of the

MOUTH model, the proliferation of the hORFs was such that their replication markedly exceeded that of the keratinocytes, and significantly reduced the patency of the channels due to transmembrane migration. This is not wholly unexpected as gingival fibroblasts have a distinct phenotype compared to other fibroblasts, especially those derived from skin[46]. The presence of hDF did not impact TEER in triculture in the early time points tested, but the IF indicated differences in tissue structure between the two. Furthermore, hDFs seeded in Transwells® resulted in pronounced piling and increased surface topography compared to hGFs.

A plateau TEER value of around $300\,\Omega\,cm^2$ is similar to other reports of primary gingival tissues grown in vitro[22] and other epithelial barrier tissues grown in submerged conditions in vitro[47–49]. Some models of oral tissue grown in ALI have reported TEER exceeding $1000\,\Omega\,cm^2$ [50], but it is unclear how physiologically relevant these numbers are, given that the barrier function of in vivo gingival tissue has never been measured[51]. We emphasize the importance of characterizing a homeostatic barrier function while noting trends in function through healthy and disease states of the MOUTH model. Although trends were consistent (i.e., number of layers, the kinetics of barrier formation), overall values of TEER and cytokine expression exhibited slight levels of variation between experiments. This variation may have been due to small variations in seeding density, hMVEC monolayer integrity, or hMVEC/hGF co-culture interaction. Although only a single hOK donor was evaluated here, it is reasonable to believe that these variations will be natural among donors[52] and, as such, provide a more physiologically-relevant platform to study states of healthy and diseased gum tissue. However, the consistent trends in barrier function over 30 days indicate a repeatable, long-term culture of gingival tissue in a microfluidic device. Baseline characterization of homeostatic tissue remains critical for every experiment as it enables a better understanding of tissue response to inflammation and subsequent treatment regimens, conditions we explore throughout this manuscript.

The long-term culture and high-throughput nature of MOUTH enables the evaluation of a large parameter space to assess various therapeutic dosing schemes, including multiple doses or early and late doses, as demonstrated here. Our studies show that barrier function decreases while release of PGE2, and inflammatory cytokine/chemokine secretion increases in response to inflammatory stimuli, trends which are largely independent of single vs. double or early vs. late dosing schemes. The kinetics of the response, as well as the magnitude of the response, was seen to differ between dosing schemes with regard to the barrier function and PGE2 release. This could be due to a change in the homeostatic baseline of the tissue during the course of the experiment. The homeostatic baseline of the tissue may be influenced directly by (A) inflammatory stimulation-induced changes in intracellular signaling pathways or components[53,54] (e.g. cellular receptors or second messengers) or (B) inflammatory stimulation-induced tissue remodeling, which has the potential to change the total cell number of each cell type present within the tissue thereby indirectly influencing intracellular signaling pathways and protein secretion and biomolecule release profiles. For example, tissue was more amenable to recovery from both a barrier function and PGE2 release standpoint, at earlier dosing times, possibly indicating a heightened state of inflammatory propensity in tissues dosed during later stages of culture.

In several instances, we have observed disruption to the hMVEC monolayer characterized by reduced cell number and changes in cell morphology when dosed with IL-1β and TNF-α. This is not unexpected, since microvascular endothelial barrier permeability is increased in states of inflammation to help

mediate an immune response[55]. Since endothelial cells are potent secretors of PGE2[56], it is possible that the number of endothelial cells present in the model, as well as their viability, are contributing to the variability of PGE2 measured in the system, particularly following stimulation schemes. Representative images of hMVECs in devices that did and did not receive stimulation indicate that the number of hMVEC can vary across a single condition (Supplementary Fig. 6). Corresponding PGE2 collected from the bottom channel indicates that the PGE2 values are likely more correlated to cell viability than cell number, however. For example, devices treated with inflammatory stimuli (Stim) versus control devices, have significantly fewer hMVEC remaining in the channel, but more than ten times the secreted levels of PGE2. While it is understood that these data support correlations between PGE2 secretion and hMVEC presence and viability, total levels of PGE2 likely reflect a complex output of the triculture tissue under the testing conditions.

Similar to PGE2 secretion, the reduction in barrier function following the dosing of inflammatory stimuli may be explained by changes in the global structure and morphology of MOUTH tissue. Clinically-relevant, pathophysiological concentrations of IL-1β and TNF-α were used as inflammatory stimuli and served to represent measured levels of these inflammatory cytokines found in the gingival crevicular fluid of patients with aggressive or chronic periodontitis[57,58]. Elevated release of PGE2 and secretion of at least ten cytokines/chemokines in MOUTH tissue were expected in response to inflammatory stimulation by IL-1β and TNF-α given numerous published reports that these analytes are upregulated in states of gingival inflammation[59–64]. Follow-on work that incorporated immune cells and additional inflammation agents, like lipopolysaccharides (LPS) that are common in the oral mucosa/gingiva, would allow for deeper analysis of pro-inflammatory markers. This is especially true for IL-10, for example, which has been linked to TNF-α production by way of LPS[65,66], but not usually triggered by TNF-α.

Our data demonstrate that the MOUTH tissue inflammatory response can be reduced using a relevant selected inhibitor, via evaluation of PGE2 release. Secretion levels of other biomarkers were less responsive to pre-dosing with the selected inhibitor. We investigated SB203580, a small-molecule inhibitor of p38-MAPK, as a potential preventative treatment for inflammatory stimulation of MOUTH tissue, given the role of p38-MAPK as a common effector and signal transduction mediator of tissue inflammation, when evaluating biomarker PGE2[67,68]. Specifically, p38-MAPK has been implicated as a mediator of tissue inflammation involved in the PGE2 signaling pathway[59]. SB203580 suppresses two (α, β) of the four (α, β, γ, δ) known isoforms of p38-MAPK, and only isoforms α and δ have been characterized in oral tissues, thus it is not surprising that our data demonstrate reduction for PGE2 and not for other inflammatory cytokines/chemokines following pre-treatment with SB203580[68]. Our studies demonstrated that treatment with SB203580 prior to IL-1β + TNF-α stimulation (Pre-SB + stim) was able to blunt the inflammatory response of MOUTH tissue relative to tissues treated only with stimuli IL-1β + TNF-α (Stim), as quantified by PGE2 release. After the first dose of stimuli following pre-treatment with SB203580, the PGE2 profile of MOUTH tissue was able to return to baseline levels (Pre-SB + stim). MOUTH tissue responded to a second dose in the same fashion, showcasing that MOUTH tissue can be dosed multiple times for extensive, consecutive studies on the same tissue samples (Pre-SB + stim 2x). While our selected inhibitor SB203580 was able to reduce MOUTH tissue response to IL-1β and TNF-α via PGE2 release, it was not able to reduce the secretion of all inflammatory cytokines and chemokines detected in our Luminex® panels. SB203580 does not target all pathways linked to inflammation,

thereby making it more effective at reducing the production of some analytes over others. These results highlight that the MOUTH system can detect levels of rare, low-concentration, and/or challenging-to-measure analytes.

We have developed a high-throughput microfluidic organ-on-chip model of gingival tissue that recapitulates physiologically-relevant metrics such as barrier function for at least 30 days of viable culture time, and have demonstrated robust response to inflammatory stimuli and protective agents on this platform. The tri-cell model mimics the multi-layered construct of gingival fibroblasts and keratinocytes seen in vivo, and incorporates a microvascular monolayer to more closely simulate the oral-to-systemic barrier. Key features of the platform, such as TEER sensors and user-controlled pumping, allow for daily quantification of barrier function and continuous perfusion of nutrients and oxygen to the tissue. The three cell types proliferate and differentiate over ten days to form a stable barrier tissue, which maintains its integrity for an additional period of 20 days or more, providing a total culture time of approximately one month and an extended window of time for the establishment of a disease model and evaluation of therapeutic candidates. The MOUTH model responds to inflammatory stimulation through a reduction in barrier function, increased levels of transport-mediated release of PGE2, and elevated secretion of inflammatory cytokines and chemokines such as MIP-3α, IL-10, and IFN-γ. This inflammatory response is then demonstrated to be modulated using a small-molecule inhibitor that has been shown to act on a relevant inflammatory pathway known to activate upon stimulation with TNF-α + IL-1β. The extended duration culture, stable testing window, ability to measure barrier function non-destructively, and increased throughput represent new capabilities in the field of oral tissue disease research, thus providing a powerful platform to study the health and disease states of gingival tissue in vitro. Additional capabilities enabled by this technology include the investigation of longer-duration dosing and the administration of a dose-response curve. Additional potential applications include more in-depth mechanistic studies, and the incorporation of an immune-competent aspect of the model. Ultimately, this platform may enable product testing and development and the opportunity to probe the systemic link between chronic and severe gingival inflammation in other tissues in the body.

## Methods
**Cell culture and device seeding**. MOUTH triculture tissue is composed of three cell types seeded in the following order: human gingival fibroblasts (hGFs) on the top surface of the membrane, human dermal microvascular endothelial cells (hMVECs) on the bottom surface of the membrane and human oral keratinocytes (hOKs) on top of the hGF layer. Matched hOK and hGF were tested from a single donor: a 60-year-old Caucasian female. The procedures were approved by the Institutional Biosafety Committee at Draper. First, a PREDICT96 microfluidic culture plate and PREDICT96 pump were sterilized via a 12 h exposure to ethylene oxide (EtO) gas and subsequently placed under a vacuum for at least 1 week. The plate was then plasma treated for 120 s. All PREDICT96 dDevices were washed with 70% ethanol and distilled water. Then, the PREDICT96 plate was coated with 0.1 mg/mL human Collagen I in acetic acid for 1 h at 37 °C. Prior to seeding the hGFs, TEER was measured for each device for baseline readings. hGFs (Lifeline Cell Technology) were then seeded at $1 \times 10^6$ cells/mL into the top channel of each device and the plate was incubated at 37 °C for 1 h. Each port in every device was topped off with 60 μL CnT-Prime Fibroblast Medium (CELLnTEC) and the plate was placed back in the incubator overnight. Media was changed the following day. Twenty-four hours later, hMVECs (Lonza) were seeded into the bottom channel of the devices at $1.8–2 \times 10^6$ cells/mL in EGM-2 MV media (Lonza) and the plate was flipped to allow the cells to settle and spread on the underside of the membrane. The PREDICT96 plate was then incubated for 1 h at 37 °C. After the incubation, the plate was flipped back over, each port of every device was topped off with 60 μL of EGM-2 MV media, and the PREDICT96 pump was placed on the plate. The flow was initiated at 10 μL/min in both the top and bottom channels. The plate was then placed back in the incubator. Two days later, designated as Day 0, a 300 ug/mL solution of MaxGel (Sigma) was added in the top channel to provide an ECM

coating to the hGF layer and incubated for 4 h at 37 °C. The devices were not underflow during this incubation. Following an incubation time of 4 h, the Maxgel solution was aspirated from the surface of the fibroblast, providing only a thin coating, which serves to smoothen out the surface topography of the fibroblast monolayer. Then, hOKs (Lifeline Cell Technology) were seeded into the top channel of each device at $1.6 \times 10^6$ cells/mL in MOUTH Custom Low Calcium Media and the plate was incubated at 37 °C for 1 h. The devices were then topped off with 60 μL MOUTH Custom Low Calcium Media, the flow was re-initiated and the plate was placed back in the incubator. About 24 h after the hOK seed, designated as Day 1, the media was changed to MOUTH Custom High Calcium Media at 75 μL per port in both top and bottom channels and maintained in the MOUTH Custom High Calcium Media thereafter. Media was changed on Day 3. On Day 5, barrier function was assessed via TEER and the media was changed. Then, TEER was read and media was changed every day for the duration of the culture. MOUTH tissue normally reached a TEER plateau around days 9–11.

**Microfluidic pumping**. The PREDICT96 pump, as described previously[29] and illustrated in Fig. 1, contains 192 individual micropumps that supply unidirectional, recirculating fluid flow to each top and each bottom channel of all 96 devices on the plate. The pumps are actuated via pneumatic lines to maintain a set stroke frequency. One stroke moves a calibrated volume of ~1.2 μl from the inlet of the channel to the outlet. The resulting hydrostatic pressure gradient drives fluid flow at a calculated rate through the microfluidic channels. To prepare the pump for experiments, an EtO-sterilized pump was removed from the sterilization bag in the biological safety cabinet and placed on a reservoir pre-filled with distilled water. The pump was primed with distilled water followed by PBS for at least 5 min each. The pump was then pumped dry by aspirating all fluid from the reservoir and placed on the microfluidic culture plate. Fluid flow in the channels was initiated after the hMVECs were attached to the membrane. The flow was initiated at 10 μL/min in both the top and bottom channels. The flow was stopped when the culture was incubating in MaxGel in preparation for hOK seeding. After hOK seeding and attachment, the flow was resumed at 10 μL/min in both the top and bottom channels. This flow rate was maintained for the rest of the culture.

**Inflammatory stimulation**. Inflammation was induced in MOUTH by administering an inflammatory stimulant cocktail consisting of 300 ng/mL of both IL-1β and TNF-α (R&D Systems) to the top and bottom channels for 24 h. TEER was measured immediately prior to inflammatory stimulation ("0 h") and every day after stimulation to quantify changes in barrier function. Media was collected after each TEER measurement by removing 120 μl of media from each top and bottom channel with a multichannel pipette. Collected media was analyzed via ELISAs or Luminex® panels (R&D Systems) to quantify PGE2 secretion as well as other inflammatory cytokine expressions. All TEER trends reported here are normalized to the baseline value measured prior to stimulation.

**TEER measurement and normalization**. TEER was measured using an EVOM² Epithelial Volt/Ohm Meter (World Precision Instruments). Blanks were measured in each PREDICT96 device prior to seeding with cells by adding 100 μL of media to each channel. Reported TEER values were calculated by subtracting each device's individual blank TEER value from that device's tissue TEER measurement and multiplied by the surface area of the overlap area. Once the tissue reached a plateau around days 9–11, baseline TEER was measured prior to treatment. The TEER of each device was normalized by dividing by the baseline. If the devices were treated twice, then they were normalized to the value measured immediately prior to treatment.

**Inhibition and inflammation prevention**. A small-molecule inhibitor, SB203580 (LC Laboratories, S-3400), was used in combination with inflammatory stimulation to assess its preventative effect on tissue response to IL-1β and TNF-α. 100 μM of the inhibitor in custom media was introduced in both the top and bottom channels of the MOUTH devices and incubated for 1–2 h. The inhibitor solution was then removed and replaced with fresh media or media containing inflammatory stimuli (see prior methods section). For devices that were treated only with SB203580, TEER was measured and media was collected prior to dosing other devices with IL-1β and TNF-α. 24 h after dosing, TEER was measured every day and media was collected every day for 48 h.

**Inflammatory biomarker analysis**. ELISAs and multiplexed Luminex® kits were used to analyze the inflammatory biomarker profiles of MOUTH tissue. PGE2 ELISA kits were run on media collected from MOUTH tissue devices. Apical media was diluted 1:18 and basal media was diluted 1:7 for the assay. During the analysis, any samples that had undetectable levels of PGE2 or levels below the ELISA kit's limit of detection (41.4 pg/mL) were set to the limit of detection value. All PGE2 data presented in this report is of bottom channel secreted PGE2 levels. In some cases, PGE2 remained undetectable for the 48 h sampling window. To explore cytokine and chemokine biomarker profiles of MOUTH tissue, custom Human Magnetic Luminex® Performance Assay kits (R&D Systems) containing a pre-mix of 18–20 analytes of interest were used to examine secreted soluble factors present in non-diluted basal media collected from devices. Luminex® kits used to

process media collected from experiments stimulated with inflammatory cytokines IL-1β + TNF-α included analytes: MCP-1, MIP-3α, Fractalkine, GROα, IP-10, G-CSF, GM-CSF, IL-4, IL-6, IL-8, IL-10, IL-17a, IL-33, PDGF-aa, PDGF-ab/bb, RANTES, VEGF, and IFN-γ. During the analyses, any samples that had undetectable levels of a specific analyte or levels below the Luminex kit's limit of detection for a given analyte were set to that analyte's lower limit of detection value, whereas any sample that had levels of a specific analyte, that exceeded the Luminex kit's limit of detection for a given analyte were set to that analytes' upper limit of detection value. The latter occurred for IL-8, and should be considered when reviewing IL-8 secretion profiles. Undiluted media samples collected from devices were run according to the manufacturer's protocol and analyzed using Luminex FLEXMAP 3D. The data collected were used to generate standard curves for each analytes using a 4- or 5-parameter logistic curve fit to determine the concentration of each analyte in the sample.

**Immunofluorescent imaging**. MOUTH tissue was fluorescently labeled with antibodies directed against specific markers for differentiated keratinocytes, cytoskeletal structures, and microvascular endothelial cells. First, cells in the devices were rinsed gently with PBS. For each rinsing, fixing, permeabilizing, and labeling step, 80–20 µL of the solution was pipetted into each port of a PREDICT96 device and the plate was rocked to initiate mixing within the channel. Cells were fixed with a 3.7% paraformaldehyde solution for 20 min at room temperature. Devices were rinsed with PBS three times and permeabilized with a 0.3% triton-x solution for 60 min at room temperature. Devices were again rinsed three times with PBS and blocked with a 3% solution of normal goat serum (NGS) for 3 h at room temperature at 4 °C overnight. Primary antibody solutions were made in a 3% NGS solution and incubated in both channels at 4 °C overnight. Mouse anti-ECAD (Abcam, AB1416) at a 1:300 dilution was used to label cell-cell junctions, rabbit anti-vWF (Abcam, AB9378) at a 1:250 dilution was used to label microvascular endothelial cells, and rabbit anti-Ck14 (Abcam, AB51054) at a 1:300 dilution was used to label the basal layer of hOKs. All channels were rinsed three times with PBS. Secondary antibodies were also mixed in a 3% NGS solution with a 1:1000 dilution of Hoechst (Invitrogen, H3570), a 1:500 dilution of Phalloidin-iFluor 633 Reagent (Abcam, ab176758) to label the actin cytoskeleton, a 1:250 dilution of goat anti-rabbit IgG Alexa Fluor™ 568, and a 1:250 dilution of goat anti-mouse Alexa Fluor™ 488 and incubated in both channels for a minimum of 1 h at room temperature. A final rinsing step was done with PBS three times before samples were ready to image. All imaging was done on an LSM 700 confocal system with Zen Black software.

**Statistics and reproducibility**. Data were evaluated in PRISM (GraphPad, Version 9.3.00). All datasets were tested for normality and homoscedasticity. Datasets that met ANOVA assumptions were tested across treatment types and time points and analyzed by post hoc Tukey tests. Datasets that did not meet ANOVA assumptions were analyzed by either the post hoc Dunnett's T3 multiple comparisons test or unpaired Welch's $t$-test, as noted in the figure captions. Statistical significance is denoted as follows: $*p < 0.0332$, $**p < 0.0021$, $***p < 0.0002$, $****p < 0.0001$. Outliers were identified using the interquartile range and removed from the dataset. Datasets represent at least three replicates, or individual devices, over at least one experiment. In some cases, as noted in the captions, datasets represent an average of three experiments with up to 12 replicates per experiment.

**Reporting summary**. Further information on research design is available in the Nature Portfolio Reporting Summary linked to this article.

## Data availability

All other data supporting the findings of this study are available from the corresponding author on request.

## Material availability

MOUTH custom media is Draper proprietary material and is available upon request by contacting the corresponding author.

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

## Acknowledgements

The authors are deeply grateful to the following Draper colleagues: Yazmin Obi and Jason Bahou for the fabrication and assembly of MOUTH plates and Tim Petrie for project planning, strategic input, and expertise. We are also very thankful to Latonya Kilpatrick, Jim Masters, and Harsh Trivedi for their contributions to scientific discussions. We thank photographers from the Draper Strategic Communication team for assistance with Fig. 1a, b. We acknowledge financial support from Colgate-Palmolive for this work.

## Author contributions

A.L.G., R.J.L., R.M., M.H.C., and E.M.V. collectively designed and performed experiments with support on experimental design and strategy from J.O. and J.L.C. A.L.G., R.J.L., and E.M.V. analyzed data. H.A., B.C.I., and B.P.C. designed and fabricated platform technology and supported its use throughout experiments. A.L.G., R.J.L., J.T.B., J.O., J.L.C., and E.M.V. contributed to the interpretation of results and writing the manuscript. All co-authors contributed to the paper revision.

## Competing interests

Work in this paper was funded by Colgate-Palmolive Company. Members of Colgate-Palmolive contributed to the experimental strategy, decision to publish, and preparation of the manuscript. The authors declare no competing interests.
