## [Peer Review File · Communications Biology]

Reviewers' comments:

Reviewer #1 (Remarks to the Author):

This manuscript describes the fabrication, characterization and testing of a high-throughput gingival triculture to model inflammatory processes. Initial characterization was through TEER, immunofluorescence microscopy of cell markers, cell and layer morphology/confluence, and ELISA/Luminex assays for baseline inflammatory markers. Stimulation with TNF-alpha and IL-1beta at several timepoints, for 24 h, with and without p38 MAPK inhibition, was performed with repeated assessments of inflammatory soluble factors.

Microfluidic tri-cultures stable over 28 days with high throughput are not common, and only several studies have developed gingival mucosal constructs in microfluidics chips. This goal is enabled by the use of the Draper P96 microfluidic plate, that uses 96 individual pumps. TEER, markers, and calcein staining of live cells seems to indicate stable and live cells through the entire 4 week period. Inflammatory signals appeared to be reliably induced in the chip, modulated by a kinase inhibitor. Some details are missing that reduce impact of the manuscript and make interpretation of some results difficult. The baseline inflammatory signals appear to be stable from day 11 to 32, but it is not clear whether the baseline is similar to healthy normal gingiva.

1. line 73, etc. Cells. Cells were stated to be primary and donor-matched. Can the authors describe in detail how many donors were tested, and at least the age range of the donors? I would expect large donor-to-donor variability in terms of baseline inflammatory signals. Was this found to be the case?
2. line 86: line I. Is this a typo?
3. Figure 1. Can the pump be displayed? 1B. Gray and red signals should be specified. Are the 96 chambers each independent (not fluidically connected, or by pressure)? What is the pore size and pore density in the membrane? The lines connecting D to E to F are confusing. Especially because the signals are different (calcein in E; VWF stain in F).
4. Figure S1. Baseline inflammatory signals are presented. Are they physiological? Please check the caption for this figure.
5. Fibroblast configuration. Can the layer order be made clear? Is it: keratinocytes, MaxGel, fibroblasts, membrane coated with collagen I, endothelial cells? Why were the fibroblasts in a monolayer rather than dispersed through the MaxGel? Since MaxGel mimics basement membrane, where is the subepithelial interstitial layer? What is the MaxGel concentration (total protein wet weight/vol)?
6. Statistics. 2-factor ANOVA and Tukey tests were reported. Were ANOVA assumptions tested, met? Especially the homoscedasticity? What factors were tested (time, treatment)? Were there any interactive effects? Please state why the F-statistic and ANOVA results were not presented--the bars and stars in figures appear to be Tukey test, pairwise comparisons only. Please check the extent of the horizontal bars and specify which groups the bars indicate.
7. 300 ng/ml of TNF-alpha and IL-1beta seem very high. What are the pathophysiological levels in vivo? Can these levels be justified? Similarly, how was the velocity of flow chosen to be 10 ul/min? Is that physiological?
8. Can the clinical/pre-clinical applicability and long-term functionality of the constructs be discussed further, highlighting strengths and weaknesses, or gaps in knowledge?

Reviewer #2 (Remarks to the Author):

In this paper, the authors used a commercial microfluidic device (Predict96 – Draper) to establish a tri-culture model of human cells (cell lines of oral keratinocytes, gingival fibroblasts, and microvascular endothelial cells) for up to 28 days to serve as a high-throughput platform to investigate cell

inflammatory responses and drug testing. Tissue responses were measured using transepithelial electrical resistance (TEER), prostaglandin E2 (ELISA), and inflammatory biomarkers (Luminex). In conclusion, the authors stated that they have developed a high-throughput microfluidic organ-on-chip model of gingival tissue that recapitulates physiologically relevant metrics such as barrier function for at least 30 days of viable culture time have demonstrated robust response to inflammatory stimuli and protective agents on this platform.

The paper has several strengths, such as (i) the relevance of the topic, since there is a shortage of organs of a chip for dental and craniofacial research, (ii) the capability to test drugs in high-throughput using TEER, and (iii) for 30 days, which are all very desirable characteristics for a system that aims to be used in large scale and to personalize drug tests.

Though, two main concerns stand out. The first one is the lack of clear evidence that this system is actually gingival and not oral mucosa. The gingival epithelium consists of three regions: oral gingival epithelium (part in contact with the oral environment), sulcular epithelium (transition facing the tooth surface), and junctional epithelium (a thin epithelial layer that tightly attaches the gingiva with the tooth, sealing the circumference of the periodontal ligament against the oral environment). Due to the lack of cementum, the system could reproduce the oral gingival epithelium, which is fine and relevant. However, the oral gingival epithelium is characterized by dense irregular epithelial ridges projecting into an adjacent fibrous connective tissue with bundles of collagen fibers that keep the tissue in place during mastication, as opposed to the oral mucosa, in which the epithelium has a flat interface with the connective tissue (as shown in Figure 1). This particular configuration of the epithelial/connective tissue interface is key to replicating a gingiva's key morphofunctional features. Other than the presence of a gingival fibroblast cell line, there is no further evidence that differentiates this system from a microphysiological oral mucosa device. As it is characterized, this platform has more oral mucosa characteristics than gingival tissue.

The second concern is about cell responses detected with the cytokine panel. It is intriguing that after stimulation with such a massive dose of TNF and IL1b, out of 18-20 cytokines and chemokines, only three were shown in this paper, being one of those IL10, which is a known anti-inflammatory cytokine related to a Th2 response (that is usually not triggered by TNF α). As this data is presented and discussed, this may raise a red flag that the system is not being able to emulate the actual physiology of the oral mucosa (or gingiva). The paper would greatly benefit from a deeper discussion that backs up these findings.

Minor notes:

- The introduction is misleading because the rationale is built upon the progression of gingivitis to periodontitis and the potential link to systemic health and diseases; however, as it is, the system cannot answer most of the problems raised in the introduction because it lacks the periodontal ligament and bone. Therefore, these statements need to be toned down throughout the text.
- The authors prioritized the characterization of the epithelium, and no description of the connective tissue was presented. For example, it would be valuable to have immunofluorescence showing the actual proportion of gingival fibroblasts relative to the other cells or evidence that the keratinocytes and endothelial cells would have behaved differently if they were co-cultured with regular fibroblasts.
- Page 3 first paragraph - PGE2 is not a hormone, but an eicosanoid, generated by cyclooxygenase 2 (COX 2) conversion of arachidonic acid.
- In figures 2 and 3, the vehicle (0.1% BSA) seems to have increased the production of PGE2 more than the early stimulus with TNF and IL1, i.e., in figure 2, the vehicle induced an output of 1000 pg/mL of PGE2, more than 10x than the presence of TNF and IL1b. It is puzzling to observe that in some conditions, BSA is more inflammatory than a cocktail of TNF and IL1b.
- Endothelial cells are a potent secretor of PGE2; maybe the variability in the PGE2 secretion could be related to the number of endothelial cells in the system. The paper would benefit from additional data correlating endothelial cell death (necrosis or apoptosis) to PGE2 using the different stimulation times. This data could be used to refine the system further and decrease the variability of the findings.
- Some early stim graphs are missing the error bar.

REVIEWER 1

Reviewer comment 1

line 73, etc. Cells. Cells were stated to be primary and donor-matched. Can the authors describe in detail how many donors were tested, and at least the age range of the donors? I would expect large donor-to-donor variability in terms of baseline inflammatory signals. Was this found to be the case?

Author Response:

We thank the reviewer for allowing us to clarify this point. We tested one donor in the scope of this work: a 60 yr old Caucasian female, in which the hOK and hGF were matched. We have clarified these details in both the methods and discussion sections (**lines 305-306, 403**). We agree with the reviewer that donor-to-donor variability will exist in baseline inflammatory signatures and discuss this in the manuscript.

Reviewer comment 2

line 86: line I. Is this a typo?

Author Response:

We appreciate the careful assessment of the text by the Reviewer. The data referenced in line 86 was missing an associated Figure label, now Fig 1J. The Results section of the manuscript has been corrected accordingly (now **line 84**).

Reviewer comment 3

Figure 1. Can the pump be displayed? 1B. Gray and red signals should be specified. Are the 96 chambers each independent (not fluidically connected, or by pressure)? What is the pore size and pore density in the membrane? The lines connecting D to E to F are confusing. Especially because the signals are different (Calcein in E; VWF stain in F).

Author Response:

We agree with the Reviewer that the pumping can be further explained and that Figure 1 would benefit from modifications. We have added an image of the pump to Figure 1 (now subset 1B) as well as additional detail on pumping mechanism and design throughout (results: **line 70**, caption: **lines 107-108**, methods: **lines 431-435**). We thank the reviewer for pointing out the labeling in Figure 1. We have removed the call out lines and updated the caption to be more explicit about images and signals. Membrane detail and signal call-outs have been added as well. The revised Figure 1 is below.

Lines 431-435 have been added to say: “The PREDICT96 pump, as described previously²⁹ and illustrated in Fig. 1, contains 184 individual micropumps that supply unidirectional, recirculating fluid flow to each top and each bottom channel of all 96 devices on the plate. The pumps are actuated via pneumatic lines to maintain a set stroke frequency. One stroke moves a calibrated volume $\sim 1.2 \mu\text{l}$ from the inlet of the channel to the outlet. The resulting hydrostatic pressure gradient drives fluid flow at a calculated rate through the microfluidic channels.”

Reviewer comment 4

Figure S1. Baseline inflammatory signals are presented. Are they physiological? Please check the caption for this figure.

Author Response:

We thank the Reviewer for this excellent question and their careful assessment of the Figure S1 caption, which has been appropriately updated. There is substantive evidence among the clinical literature that the prevalence of the most robustly expressed analytes, MCP-1, IL-8, GRO α , and VEGF detected from our 18x-plex investigation of the MOUTH model, is consistent with observed biomarker profiles of gingival crevicular fluid (GCF) from periodontally healthy human subjects (refs 43, 44). There are a number of additional reports that seek to characterize the baseline inflammatory cytokine and chemokine profiles of gingival crevicular fluid, saliva, and/or blood serum of periodontally healthy human subjects using multiplexed fluorescent bead-based immunoassays or enzyme-linked immunosorbent assays (ELISA). Review of these publications reveals that all 18 analytes surveyed have been detected at varying levels in periodontally healthy subjects. We have summarized many of the relevant, available clinical reports in a table below. We have also edited the discussion (**lines 269-273**) for this figure.

Additional text and 15 references have been added to lines **269-273** in the discussion: **Additionally, during the 30 day culture period, we quantified the baseline inflammatory profile of the MOUTH model by multiplexed fluorescent bead-based immunoassay and observed that the prevalence of the analytes surveyed (Supplementary Fig. 1) was consistent with observed inflammatory cytokine and chemokine profiles of gingival cervical fluid from periodontally healthy human subjects³¹⁻⁴⁴**

Cytokines / Chemokines	MOUTH model concentration (pg/mL) \pm SD D11	MOUTH model concentration (pg/mL) \pm SD D32	Healthy human clinical specimen concentration (pg/mL) \pm SD or SEM by Luminex or ELISA	Literature

MCP-1 (CCL2)	2,173 ± 28	840 ± 365	19,700 ± 3,270 (SD) 13,180 ± 1,120 (SEM)* 0.82 ± 1.11 (SD) pg/30-s sample**	a, b, c
MIP-3a	177 ± 22	300 ± 63	~5 ± 12.5 (SD) 0.96 ± 1.78 (SD) pg/30-s sample**	c, d
Fractalkine (CX3CL1)	396 ± 12	365 ± 20	3.87 ± 3.02 (SD) pg/30-s sample**	c
GROα (CXCL1)	1,112 ± 157	368 ± 35	~250 ± 250 (SD) 160.42 ± 94.21 (SD) pg/30-s sample**	c, e
IL-10	2 ± 0	3 ± 1	2,400 ± 1,200 (SEM)*	b
IL-8	2,407 ± 518	1,495 ± 301	2,454,710 ± 1,150 (SEM)* 206.13 ± 46.63 (SD) ~ 250 ± 125 (SD) 170.98 ± 176.96 (SD) pg/30-s sample**	b, c, e, f
G-CSF	465 ± 108	438 ± 38	77,620 ± 1,320 (SEM)*	b
GM-CSF	154 ± 59	63 ± 5	2,400 ± 1,200 (SEM)* 91.31 ± 10.41 (SD) 0.67 ± 0.76 (SD) pg/30-s sample**	b, c, g
IFNγ	8 ± 1	8 ± 0	1,550 ± 1,200 (SEM)* ~ 0.28 ± 0.16 (SD) 0.92 ± 0.54 (SD) pg/30-s sample**	b, c, d
IL-10	19 ± 1	17 ± 1	2.41 ± 0.57 (SD) ~ 0.475 ± 0.188 (SD) 0.69 ± 0.32 (SD) pg/30-s sample**	c, d, f
IL-17a	2 ± 0	2 ± 0	15 ± 2 (SD) 22.81 ± 23.63 (SD) 1,200 ± 1,100 (SEM)*	b, h, i
IL-33	14 ± 2	12 ± 1	414.36 ± 49.461 (SD) 8,300,000 ± 4,857,000 (SD)	j, k
IL-4	1 ± 0	1 ± 0	5,250 ± 1,350 (SEM)* ~ 0.42 ± 0.103 (SD) 0.25 ± 0.21 (SD) pg/30-s sample**	b, c, d
IL-6	175 ± 62	19 ± 5	1,580 ± 1,170 (SEM)* 0.43 ± 0.22 (SD) ~10 ± 5 (SD) 3.37 ± 8.47 (SD) pg/30-s sample**	b, c, e, f
PDGF _{aa}	172 ± 22	167 ± 22	~ 0.0088 ± 0.0088 (SD) pg/ug protein***	i
PDGF _{ab/bb}	21 ± 2	21 ± 4	~ 0.5 ± 0.038 (SD) pg/ug protein***	i

RANTES	94 ± 0	95 ± 0	4,900 ± 1,380 (SEM)*	b
VEGF	813 ± 65	111 ± 19	31.9144 ± 3.06842 (SD) 134,890 ± 1150 (SEM)*	b, m

*Reported units (log mean ng/mL) converted to pg/mL.

**Reported units are pg/30-s sample; provided for reference or when pg/mL data is unavailable

*** Reported units are pg/ug protein; provided for reference or when pg/mL data is unavailable

a Pradeep, et al., Arch Oral Biol, 2009. doi: 10.1016/j.archoralbio.2009.02.007

b Offenbacher, et al., J Clin Periodontol, 2010. doi: 10.1111/j.1600-051X.2010.01543.x

c Bamashmous, et al., Front Oral Health, 2021. doi: 10.3389/froh.2021.689475

d Zhang, et al., PLoS One, 2021. doi: 10.1371/journal.pone.0244806

e Roy, et al., Front Oral Health, 2022. doi: 10.3389/froh.2021.815728

f Escalona, et al. Invest Clin., 2016. PMID: 28429894

g Liao, et al., African J of Biotech, 2011. doi: 10.5897/AJB10.2252

h Jimenez, et al. Anais Brasileiros de Derm. doi: 10.1016/j.abd.2020.08.008

i Sadehi, et al., Cent Eur J Immunol, 2018. doi: 10.5114/ceji.2018.74876

j Pai B and Paradeep, Bull Tokyo Dent Coll, 2019. doi: 10.2209/tdcpublish.2019-0002

k Buduneli, et al. J Periodontol, 2012. doi: 10.1902/jop.2011.110239.

l Villa, et al. Sci Reports, 2016. doi: 10.1038/srep23060

m Padma, et al, J Clin Diagn Res, 2014. doi: 10.7860/JCDR/2014/8450.5163

Reviewer comment 5

Fibroblast configuration. Can the layer order be made clear? Is it: keratinocytes, MaxGel, fibroblasts, membrane coated with collagen I, endothelial cells? Why were the fibroblasts in a monolayer rather than dispersed through the MaxGel? Since MaxGel mimics basement membrane, where is the subepithelial interstitial layer? What is the MaxGel concentration (total protein wet weight/vol)?

Author Response:

The reviewer is correct in their assessment of the configuration of the layers of the tissue model. The order from top to bottom is: keratinocytes, Maxgel, fibroblasts, coated membrane, endothelial cells. While Maxgel is most often utilized as a 3D scaffold, it was not used as such in the configuration of this model. The microfluidic nature of the PREDICT96 environment does not allow for a 3D full-thickness gel embedded with fibroblasts, as creation of such would occlude the microfluidic channel, thus preventing the incorporation of additional cell types as well as fluid flow – both of which are demonstrated in this model. For this reason, Maxgel was used as an ECM coating rather than a 3D scaffold. A 300 ug/mL Maxgel solution was added to provide an ECM coating to the fibroblast layer in order to improve keratinocyte attachment and integration into the model. Following an incubation time of 4 hours, the Maxgel solution was aspirated from the surface of the fibroblasts, providing only a thin coating, which serves to smoothen out the surface topography of the fibroblast monolayer. Both seeding order and Maxgel details have been clarified **throughout the text**.

The first few lines of the methods section (**lines 400-403**) now read: “MOUTH tri-culture tissue is composed of three cell types seeded in the following order: human gingival fibroblasts (hGFs) on the top surface of the membrane, human dermal microvascular endothelial cells (hMVECs) on the bottom surface of the membrane and human oral keratinocytes (hOKs) on top of the hGF layer. Matched hOK and hGF were tested from a single donor: a 60 yr old Caucasian female.”

Further in the section in **lines 417-421**, we have added content to read: “Two days later, designated as Day 0, a 300 ug/mL solution of MaxGel (Sigma) was added in the top channel to provide an ECM coating to the hGF layer and incubated for 4 hours at 37°C. The devices were not under flow during this incubation. Following an incubation time of 4 hour, the Maxgel solution was aspirated from the surface of the fibroblast, providing only a thin coating, which serves to smoothen out the surface topography of the fibroblast monolayer.”

Reviewer comment 6

Statistics. 2-factor ANOVA and Tukey tests were reported. Were ANOVA assumptions tested, met? Especially the homoscedasticity? What factors were tested (time, treatment)? Were there any interactive effects? Please state why the F-statistic and ANOVA results were not presented--the bars and stars in figures appear to be Tukey test, pairwise comparisons only. Please check the extent of the horizontal bars and specify which groups the bars indicate.

Author Response:

We appreciate the opportunity to improve and clarify the methodology of our statistical testing. A close evaluation of our statistical analysis revealed that not all ANOVA assumptions had been tested. Now, all datasets have been tested for normality and homoscedasticity after being log transformed using PRISM 9.3.0. The log transformation was done to bring datasets onto the same scale. Datasets that met ANOVA assumptions were tested across time point and treatment (example: Stim, vehicle control, SB) and analyzed by post-hoc Tukey tests. Datasets that did not meet ANOVA assumptions were analyzed by either the post-hoc Dunnett’s T3 multiple comparisons test or unpaired Welch’s t-test. All bar graphs are now in dot-plot format to show data distribution. The captions of every figure comprised of a data plot contains details on statistical analysis and replicate counts.

Reviewer comment 7

300 ng/ml of TNF-alpha and IL-1beta seem very high. What are the pathophysiological levels *in vivo*? Can these levels be justified? Similarly, how was the velocity of flow chosen to be 10 ul/min? Is that physiological?

Author Response:

We thank the Reviewer for the astute questions and appreciate the opportunity to answer each one. The levels of IL-1beta and TNF-alpha stimulus studied in this manuscript are similar to *in vivo* concentration ranges measured in cases of severe, chronic periodontitis. Pathophysiological levels of TNF-alpha and IL-1beta have been measured by ELISA in gingival crevicular fluid between 0.0001 – 1,030 ng/mL and 67 – 538 ng/mL, respectively, in patients with aggressive or chronic periodontitis (refs 57,58). We have edited the discussion of the manuscript to reflect inclusion of these references (**lines 343-347**).

Regarding the hemodynamic fluid shear stress (FSS) level selected, 10 uL/min is equivalent to 0.1 dynes/cm², which provides waste removal, nutrient recirculation and oxygenation of the tissue. The endothelial compartment or vascular network of *in vivo* human tissues can be subjected to FSS levels ranging from <1 dyne/cm² to > 600 dynes/cm² depending on the tissue type and region of the vascular network under consideration^{m,n}. Although beyond the scope of this body of work, it would be interesting

and useful to determine the optimal FSS profile for this model by performing studies of the model cultured at different FSS levels.

m, B. J. Ballermann, A. Dardik, E. Eng and A. Liu, *Kidney Int.*, 1998, **54**, S100–S108.

n, N. Baeyens, S. Nicoli, B. G. Coon, T. D. Ross, K. Van den Dries, J. Han, H. M. Lauridsen, C. O. Mejean, A. Eichmann, J.-L. Thomas, J. D. Humphrey and M. A. Schwartz, *eLife*, , DOI:10.7554/eLife.04645.

Reviewer comment 8

Can the clinical/pre-clinical applicability and long-term functionality of the constructs be discussed further, highlighting strengths and weaknesses, or gaps in knowledge?

Author Response:

This is a highly relevant discussion point to our scope of work and we thank the Reviewer for the opportunity to highlight it in the manuscript. The discussion has been updated to include strengths and weaknesses of pre-clinical applicability of MOUTH as well as its long-term functionality (**lines 256 – 264**). Briefly, the pre-clinical applicability of MOUTH is valuable in its current state for the oral- and self-care industries that do not use animal models to test products, or for companies who are interested in a more complex model (multiple cell types, human primary cells, physiological architecture) to study cell and tissue response to external stimuli. However, for biotech industries, the platform does not yet replace *in vivo* studies and will need more clinical validation before doing so. Long-term functionality of our MOUTH model is critical to ensure multi-week test windows, and is a huge advancement for complex tissue models, but even one month may not be adequate for more animal-based chronic disease progression studies, such as periodontitis.

The **first paragraph of the discussion** has been expanded as follows: "Validation using known inflammatory stimuli and clinical metrics of inflammatory response are crucial to establishing an *in vitro* gingival tissue model that possess clinical significance. Preclinical pipelines, in particular, rely on traditional *in vitro* cell culture or animal models, which each suffer from low predictive accuracy and low throughput, respectively. An advanced, high throughput tissue culture platform like MOUTH offers a balance of complex human tissue and throughput in the preclinical space, especially for oral- and self-care production companies that do not use animal models. Furthermore, the month-long functionality of the model provides a platform from which to better study and understand multi-window scenarios such as inflammation response and tissue recovery in the presence of therapeutics, or acute disease progression. Further validation and development of the model would be needed to replace animal models for more chronic disease and toxicity evaluation."

REVIEWER 2

Reviewer comment 1

The introduction is misleading because the rationale is built upon the progression of gingivitis to periodontitis and the potential link to systemic health and diseases; however, as it is, the system cannot answer most of the problems raised in the introduction because it lacks the periodontal ligament and bone. Therefore, these statements need to be toned down throughout the text.

Author Response:

Upon re-evaluation of the introduction, we agree with the reviewer that the motivational language of the introduction is misleading when paired with the ultimate scope of the manuscript. We have adjusted the introduction, specifically in the first paragraph (**lines 23-29**), as well as the abstract (removed content in **first two sentences**) to limit the significance to early stage gum disease and inflammation, rather than progression to periodontitis.

The **first paragraph** now reads: “Gum tissue serves as the primary protective barrier and physical defense in the mouth against bacterial invasion into gingival tissue and the blood stream. Half of American adults suffer from periodontal disease¹, and the World Health Organization reports that gum disease remains a very important global health burden². **In early stages of** gum disease, microbial plaques accumulate and form biofilms, eliciting inflammatory-mediated gingivitis, a preventable and reversible infection of the gingival tissue and alteration of the protective barrier³.. The oral disease research community remains limited by the tools and technology available to probe gum disease pathophysiology, **and its broader consequences to systemic health and disease**. This gap in laboratory research tools has hampered advances in understanding oral inflammation at a cellular level, **as well as the ability to evaluate oral therapeutic and prophylactic agents *in vitro*.**”

Reviewer comment 2

The authors prioritized the characterization of the epithelium, and no description of the connective tissue was presented. For example, it would be valuable to have immunofluorescence showing the actual proportion of gingival fibroblasts relative to the other cells or evidence that the keratinocytes and endothelial cells would have behaved differently if they were co-cultured with regular fibroblasts.

Author Response:

We thank the reviewers for the opportunity to provide an amended description of the connective tissue portion of the MOUTH model. During the development of MOUTH, we evaluated the performance of our model when cultured with human dermal fibroblasts (hDF) vs gingival fibroblasts (hGF). We also evaluated the performance of human oral fibroblasts (hORF) from a different donor. The presence of hDF did not impact TEER in triculture in the early time points tested, but the IF indicated differences in tissue structure between the two. Furthermore, hDFs seeded in Transwells have pronounced piling and surface topography compared to hGFs. In the microfluidic microenvironment of the MOUTH model, the proliferation of the hORFs was such that their replication markedly exceeded that of the keratinocytes, and significantly reduced patency of the channels due to transmembrane migration. The gingival fibroblasts, conversely, formed a dense layer prior to keratinocyte seeding, but stabilized and seemingly reduced in number by the end of 28 days post-keratinocyte seed. End-point imaging shows sparse spatial distribution of fibroblasts in close proximity to the basal layer of the epithelium. Further

description of human gingival fibroblasts growth has been added to the results text (**lines 285-296**), which now read:

“hGFs formed a confluent monolayer prior to introducing the hOK. However, after 30 days of triculture conditions, the hGF were often difficult to distinguish from the other two tissue types but could be seen sporadically throughout the basal layers due to their elongated morphology, TE-7 expression, and actin structure (Supplementary Fig. 5). Two additional primary fibroblast cell types were evaluated while establishing the MOUTH triculture model: human oral fibroblasts (hORF) and human dermal fibroblasts (hDF). In the microfluidic microenvironment of the MOUTH model, the proliferation of the hORFs was such that their replication markedly exceeded that of the keratinocytes, and significantly reduced patency of the channels due to transmembrane migration. This is not wholly unexpected as gingival fibroblasts have a distinct phenotype compared to other fibroblasts, especially those derived from skin⁴⁵. The presence of hDF did not impact TEER in triculture in the early time points tested, but the IF indicated differences in tissue structure between the two. Furthermore, hDFs seeded in Transwells resulted in pronounced piling and surface topography compared to hGFs.”

Reviewer comment 3

Page 3 first paragraph - PGE2 is not a hormone, but an eicosanoid, generated by cyclooxygenase 2 (COX 2) conversion of arachidonic acid.

Author Response:

The reviewer is correct in noting that PGE2 is an eicosanoid and is not a hormone. We have edited the manuscript text (**line 91**) to correctly reflect the characterization of this important active lipid mediator derived from arachidonic acid.

Reviewer comment 4

In Figures 2 and 3, the vehicle (0.1% BSA) seems to have increased the production of PGE2 more than the early stimulus with TNF and IL1, i.e., in figure 2, the vehicle induced an output of 1000 pg/mL of PGE2, more than 10x than the presence of TNF and IL1b. It is puzzling to observe that in some conditions, BSA is more inflammatory than a cocktail of TNF and IL1b.

Author Response:

We have investigated the possible causes for the PGE2 levels measured for the Vehicle condition at time point 0, but have not isolated a clear and identifiable cause. We hypothesize that the tissue devices, which were blindly selected, for the Vehicle condition coincidentally generated higher baseline levels of PGE2 at time point 0 relative to the devices selected for the Stim condition. It is important to clarify here that the early stimulus cytokines TNF α and IL-1b have not been added at time point 0, as this is the baseline condition prior to stimulus addition. Stimulus addition occurs immediately following the time point 0 collection and measurement. The methods section has been modified to make this more clear (**line 447**).

It now reads: “TEER was measured immediately prior to inflammatory stimulation (“0 hours”) and every day after stimulation to quantify changes in barrier function.”

Reviewer comment 5

Endothelial cells are a potent secretor of PGE2; maybe the variability in the PGE2 secretion could be related to the number of endothelial cells in the system. The paper would benefit from additional data correlating endothelial cell death (necrosis or apoptosis) to PGE2 using the different stimulation times. This data could be used to refine the system further and decrease the variability of the findings.

Author Response:

We thank the reviewer for the helpful comments regarding endothelial cells as potent secretors of PGE2. We agree with the reviewer that the endothelial cell number and viability may be influencing the PGE2 secretion profiles and have addressed this point in our revised manuscript in the discussion section (**Lines 331 – 341**). Upon extensive review of images and associated PGE2 data, we have made two observations: 1. The number of endothelial cells in un-stimulated devices is less variable than stimulated devices and 2. the viability of the hMVEC cells, as evidenced by cell number and morphology, are greatly impacted by MOUTH inflammatory conditions. We therefore hypothesize that variation in cell number and baseline viability within a condition can contribute to occasional PGE2 variability in our 32 day triculture model. Of course, the overall secretion of PGE2 is likely a complex relationship between all three cell types, especially the hGFs, and their environmental conditions. We have updated SI figure 6 with additional immunofluorescence images demonstrating 1. Consistency of hMVEC-channel PGE2 secretion in devices with good cell coverage, and 2. stark differences in cell number and morphology of hMVEC treated receiving stim conditions.

The discussion in **lines 331-341** has been expanded to summarize the above: “Since endothelial cells are potent secretors of PGE2⁵⁴, it is possible that the number of endothelial cells present in the model, as well as their viability, are contributing to the variability of PGE2 measured in the system, particularly following stimulation schemes. Representative images of hMVECs in devices that did and did not receive stimulation indicate that the number of hMVEC can vary across a single condition (Supplementary Fig. 6). Corresponding PGE-2 collected from the bottom channel indicates that the PGE-2 values are likely more correlated to cell viability than cell number, however. For example, devices treated with inflammatory stimuli (Stim) versus control devices, have significantly fewer hMVEC remaining in the channel, but more than 10 times the secreted levels of PGE2 . While it is understood that these data

support correlation between PGE2 secretion and hMVEC presence and viability, total levels of PGE2 likely reflect a complex output of the triculture tissue under the testing conditions."

Reviewer comment 6

Some early stim graphs are missing the error bar.

Author Response:

We thank the reviewer for this important observation. We have modified the manuscript Figures, i.e. **Figure 3E**, to include error bars where they were missing.

MAJOR:

Reviewer comment major 1

The first one is the lack of clear evidence that this system is actually gingival and not oral mucosa. The gingival epithelium consists of three regions: oral gingival epithelium (part in contact with the oral environment), sulcular epithelium (transition facing the tooth surface), and junctional epithelium (a thin epithelial layer that tightly attaches the gingiva with the tooth, sealing the circumference of the periodontal ligament against the oral environment). Due to the lack of cementum, the system could reproduce the oral gingival epithelium, which is fine and relevant. However, the oral gingival epithelium is characterized by dense irregular epithelial ridges projecting into an adjacent fibrous connective tissue

with bundles of collagen fibers that keep the tissue in place during mastication, as opposed to the oral mucosa, in which the epithelium has a flat interface with the connective tissue (as shown in Figure 1). This particular configuration of the epithelial/connective tissue interface is key to replicating a gingiva's key morphofunctional features. Other than the presence of a gingival fibroblast cell line, there is no further evidence that differentiates this system from a microphysiological oral mucosa device. As it is characterized, this platform has more oral mucosa characteristics than gingival tissue.

Author Response:

We thank the reviewer for their expertise in this area and drawing attention to the importance of both understanding and mimicking the native microenvironment when establishing in vitro models of tissue. As the reviewer mentions, rete ridges are an important feature of the oral gingival epithelium, particularly as they help provide the firm tissue base needed to support mastication forces. It would be of interest to us to incorporate previously developed membrane surface topography (ref 28) in future MOUTH models to mimic/support the formation of epithelial ridges at the connective tissue interface. We believe the gingivally-derived cell types in our current model support other gingival-specific histology including 3-4 layers of parakeratinized stratified squamous epithelium, especially the granular and spinous layer, as seen in Fig. 1H and 1I. To further elucidate this, a 3D confocal video has been added to the Supplementary document (Supplementary video 1) illustrating these features and highlighting some macroscale topography between the cell layers. We recognize that no in vitro model will fully replicate original tissue structure and take this under consideration while designing and evaluating said tissue models, but for the reasons above and while keeping the intended application of the model (studying states of inflammation and health), we hope to keep the labeling of the MOUTH model as gingival epithelium rather than oral mucosa. We have addressed the importance of this comment in the discussion section, **lines 276-284**:

“A video of MOUTH tissue structure (Supplementary video) demonstrates differentiated morphology as the optical field of view moves from the densely packed basal layer of cells through multiple layers of parakeratinized stratified squamous epithelium with evidence of granular and spinous layers. In addition to the gingival- origin of the keratinocytes and fibroblasts, the tissue morphology indicates gingival-specific histology relevant to the applications of MOUTH. Additional features, including rete ridges involved supporting mastication forces, would be of interest to model in future MOUTH applications using either previously developed membrane surface topography or other engineered microenvironmental features.”

Reviewer comment Major 2

It is intriguing that after stimulation with such a massive dose of TNF and IL1b, out of 18-20 cytokines and chemokines, only three were shown in this paper, being one of those IL10, which is a known anti-inflammatory cytokine related to a Th2 response (that is usually not triggered by TNFa). As this data is presented and discussed, this may raise a red flag that the system is not being able to emulate the actual physiology of the oral mucosa (or gingiva). The paper would greatly benefit from a deeper discussion that backs up these findings.

Author Response:

We thank the reviewer for mentioning this point. We had fully intended to include the full dataset from the Luminex panel and have corrected that mistake by including them in the supplemental document (Supplement figs 7 and 8), along with a deeper discussion into the findings. We have also removed the former Supplement fig. 8, which included IL-6 and IL-8 data, since it is now redundant with the full panel. Briefly, Fractalkine, GM-CSF, G-CSF, RANTES, VEGF, IL-33, IP-10, and IL-6 were all upregulated following inflammatory stimulation, a result that was inhibited when pre-treated with SB by within at least 48 hours. Most of these factors are secreted by endothelial cells and are all implicated in immune response to insult. An upregulation of IFN γ could be triggering the IP-10 response from the system. GRO α , IL-8, and IL-4 were also upregulated following inflammation stimulation, but not protected when pre-treated with SB. MCP-1, G-CSF, PDGF $\alpha\alpha$, and PDGF $\beta\beta$, and expression levels were not significantly affected by IL-1 β /TNF α 1B/TNF α stimulation. Results have been updated to reference the data accordingly in **lines 129-133** as well as **lines 161-163** and **lines 188-191**. Furthermore, follow-on work that incorporated immune cells that could respond to these upregulated cytokines/chemokines would allow for deeper analysis of the interaction between the immune and tissue response to inflammatory stimulation.

In reference to IL-10 response in our model, we have found reference to IL-10 secretion being linked to TNF α production by way of inflammatory agents like lipopolysaccharides (LPS), that are common in the oral mucosa/gingiva. There have been several studies conducted on the effect of IL-10 on LPS-induced inflammatory responses (refs 65,66). LPS can trigger both TNF α and IL-10 production. IL-10 has been shown to mediate TNF α levels. In mice, neutralization of TNF α resulted in significant reduction of LPS-inducible IL-10 production, suggesting that TNF α expression could affect IL-10 production. The relationship between TNF α and IL-10 should be further studied to better understand LPS-induced inflammatory response. Discussion on this is included in **lines 349-352**.

REVIEWERS' COMMENTS:

Reviewer #1 (Remarks to the Author):

All comments addressed sufficiently.

Reviewer #2 (Remarks to the Author):

I was pleased with the way that authors comprehensively addressed the questions and amended the manuscript. I am happy to recommend it for publication as is.